# Effects of Prolonged Storage Condition on the Physicochemical and Microbiological Quality of Sachet Water and Its Health Implications: A Case Study of Selected Water Brands Sold within Samaru Community, Northwest Nigeria

**Taiwo Adekanmi Adesakin** [1,*]**, Abayomi Tolulope Oyewale** [2]**, Ndagi Abubakar Mohammed** [1]**, Umar Bayero** [1]**, Adebukola Adenike Adedeji** [3]**, Idowu Adedeji Aduwo** [3]**, Adetolani Christianah Bolade** [4] **and Maryam Adam** [1]

[1]  Department of Biology, Ahmadu Bello University, Zaria 810107, Nigeria
[2]  Institute of Ecology and Environmental Studies, Faculty of Science, Obafemi Awolowo University, Ile-Ife 220212, Nigeria
[3]  Department of Zoology, Obafemi Awolowo University, Ile-Ife 220212, Nigeria
[4]  School of Health Information Management, OAUTH, Ile-Ife 220212, Nigeria
*  Correspondence: ataiwonelson@gmail.com; Tel.: +234-8036560348

**Abstract:** The aim of this research is to investigate the effect of long-term storage conditions on the physicochemical and microbial quality of selected sachet water brands sold within the Samaru community and its health implication for consumers. Three brands of sachet water were subjected to different storage conditions and were analyzed for microbial and physicochemical parameters at intervals of 3 weeks for a period of 3 months, based on procedures and standard methods of APHA (2005). The highest pH means concentration was recorded in sachet water samples used for control (7.14 ± 0.24) while EC, TDS, BOD, and calcium (336.67 ± 73.69 μS/cm, 168.33 ± 34.89 mg/L, 1.87 ± 0.39 mg/L and 2.97 ± 0.19 mg/L) were recorded in the sachet water samples stored on the floor. A total number of 15 fungal species and 4 bacteria species were identified from the three selected brands of sachet water examined. *Aspergillus niger* and *Penicillium* spp. have the highest species occurrence of 25% among the fungus identified while *Proteus* Sp. recorded the highest species occurrence (50%) among the bacteria isolated. Sachet water samples stored on the floor have the highest abundances of microbial species of five and six. All the physicochemical parameters were within the stipulated limits set by the World Health Organization and the Nigeria Standard for Drinking Water Quality, but fail microbial tests due to the presence of pathogens such as *Aspergillus*, *Candida*, *Vibro*, *Yersinia*, etc., that can cause a wide range of life-threatening system infections in a patient with mild immune-deficiency. Being under prolonged storage and the type of storage conditions can encourage the regrowth of microbial in packaged water under favorable environmental conditions, to levels that may be harmful to humans.

**Keywords:** pathogens; harmful; monitoring; microbial; water storage

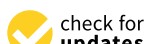



## 1. Introduction

Water is the most prevalent naturally occurring substance and a necessary resource for the survival of all life forms on Earth. However, most people lack adequate safe drinking water due to the increase in the human population, coupled with anthropogenic activities that pollute drinking water sources [1]. The unavailability of good quality drinking water around the globe has claimed more lives through water-related diseases than any war occurrence through guns among the people living in under-developed nations around the world [2]. Mortality in children under five years of age from water-related diseases annually is estimated to be approximately 4 million in developing nations, while 2.3 billion people worldwide have mortality and morbidity associated with water-related illnesses [3]. According to [4], their statistics showed that 748 million (11%) of all the people living in

the world lack access to potable water, while in Nigeria, only 36% of the approximately 200 million populaces have access to drinkable water. The potable water must conform to the specified standard in terms of its physical, chemical, microbiological and radiological quality, while its microbiological quality is usually the most important in drinking water quality monitoring [5]. The WHO [5] defines safe drinking water as water that poses no appreciable danger to health over the course of a lifetime consumption, taking into account any potential differences in sensitivity between life phases. The provision of adequate and clean drinking water to a country's population is one of the main issues facing developing nations today, while the larger percentage of people living in African countries lack access to potable water [6,7]. In the previous four decades, Nigeria has seen a gradual fall in the quality of water delivery services provided by most public water agencies, which are often unreliable, of poor quality, and not sustainable due to an increase in the population, deterioration of the water supply facilities, poor management of resources, water pollution and a lack of commitment from the government towards proving adequate and safe drinking water to its citizens, which has resulted in a high cost of production for potable water [8]. The increase in demand for safe drinking water has led to the production of sachet water, popularly called "pure water". It is the most widely consumed drinking water by Nigerians because it is relatively cheaper and considered the most commercially affordable potable drinking water [9,10].

In Nigeria today, the easy availability of drinking water in packaged forms has led to a significant growth in the water industry, with millions of liters of these products consumed everyday by Nigerians [11]. The integrity of sachet water is doubtful; in fact, it has been reported that most of these private sachet water productions have little knowledge about manufacturing practices, do not treat their water source and do not follow the strict standards set by [12]. Occasionally, a contamination of sachet water may occur either during processing, packaging, transportation, storage usage and improper handling by hawkers or from the source of the water used for the production [13]. The microbial pathogens such as the *Salmonella*, *Shigella*, *Vibrio*, *Campylobacter*, *Yersinia*, *Cryptosporidium* and *Giardia* species has been reportedly found in package water, which can cause water-borne diseases and epidemics [14,15] such as diarrhea, cholera, dysentery, typhoid fever, legionnaire's disease and parasitic diseases, thus this is of serious concern to consumers and public health authorities. Studies on the bacteriological quality of sachet water in some African towns have documented varying degrees of contamination, despite its widespread use [5,16,17]. In Nigeria, most of the sachet water parameters fell below the WHO [12] and NSDWQ [18] standards for drinking water quality, and therefore are of doubtful quality, posing health risks to consumers [19].

Sachet or table water, although presumed safe, is often exposed to different storage conditions for several months without consideration of the possible implications of its storage on its quality [20]. Sachet water's deterioration has been influenced by the persistence and regeneration of harmful microorganisms in it as a result of long-term storage [21]. Till date, there is a dearth of information on the effect of long-term storage conditions on the microbial and physicochemical alteration of sachet water quality [22–24]. This study will provide information on the impact of storage conditions on the microbiological and physicochemical quality of the sachet water sold within the Samaru community, Zaria, northwestern Nigeria, and its health implications on the final consumers.

## 2. Materials and Methods

### 2.1. Study Area

The study areas are located within the Samaru community and its environs which include the Ahmadu Bello University campus in Samaru, Sabon Gari Local Government, Kaduna state, Nigeria. Samaru is located in the northern Guinea savannah zone of Nigeria falling within a Longitude of 7°37′60″ E and latitude of 11°10′00″ N, with an altitude of 763 m above sea level. It has a tropical climate with a well-defined rainy season, which occurs from May to October, and the dry season from November to April. Mean

monthly temperature ranges from 13.8 °C to 36.7 °C with an annual rainfall of 1090 mm are characteristic of Zaria [6].

Sample Collection

The list of sachet water brands with the National Agency for Food and Drug Administration and Control certification (NAFDAC) sold within the Samaru community was obtained and assigned numbers from 1 to 15, forming the sample frame. Five sachet water samples were purchased for each of these 15 sachet water brands sold within the Samaru community from the manufacturer and were properly labeled and transported to the laboratory for preliminary physicochemical and microbiological analysis. Three brands of sachet water selected for this research were within the World Health Organization (WHO) and Nigeria Standard for Drinking Water Quality (NSDWQ) permissible limit ranged for potable drinking water. Three brands of sachet water samples (A = TSK, B = M-Square, and C = Fahad) were collected from different parts of the Samaru community in bags within 24 h of production and stored in a room at an ambient temperature. The sub-samples were subjected to different storage conditions, namely, some of this sachet water was stored inside a fridge, on the floor, and on a wooden plank in triplicates for the physicochemical characterization and bacteriological assay using [25,26] analytical methods.

The physicochemical parameters of the sachet water were analyzed immediately after collection and subsequently repeated on 3 weekly bases for all the water samples stored in the refrigerator, on the floor, and on the wooden plank. The water samples were examined using the appropriate quality control techniques and standard methodologies within the holding times of the respective parameters. The temperature, pH, electrical conductivity, and total dissolved solids (TDS) were determined electrometrically with a multi-parameter data logger (Hanna model HI991300, Hanna Instruments, Singapore). The dissolved oxygen (DO) was determined by Winkler's Method, where glass reagent bottles of 250/125 mL capacity were gently filled with the sachet water samples and were fixed using Winkler's A (manganous sulphate solution) and Winkler's B (alkali-iodide) reagents, then conc. sulphuric acid was added to free the fixed oxygen inside the water sample and they were titrated with a sodium thiosulphate solution. The samples for (biological oxygen demand) $BOD_5$ determination were equally collected in glass reagent bottles but were not fixed. The BOD water samples were kept in a dark cupboard at room temperature (25 °C) for five days after which their oxygen content was determined by the Winkler methods, as described by [27], and these were used to determine the amount of dissolved oxygen at the end of the incubation period. Nitrate was determined using the Brucine sulfanilic acid method [28]. The chloride was analyzed by Mohr's titration method, the spectrophotometric method was adopted to analyze the phosphate while the total hardness was also determined by titrimetric method using a dropper to add an ethylenediaminetetraacetic acid (EDTA) solution to the water sample.

The sachet water was analyzed for its microbiological quality to enumerate the total heterotrophic bacteria count (on nutrient agar (NA)), the total coliform bacteria count (on eosin methylene blue (EMB) agar), and the total fungi count (on potatoes dextrose agar (PDA)), using a serial dilution method and pour plate techniques. For the microbiological coliform analysis, a membrane filter approach was used. Each sample was filtered through a filter membrane with a constant pore size of 0.45 m. The bacteria were retained on the surface of the membrane filter. The membrane filter was placed in a Petri dish containing m-Faecal coliform broth and was incubated at 44 °C for 18 to 24 h. The bacterial cultured plates were incubated at 37 °C for 24 h while the fungi plates were incubated at 25 °C for 72 h. While the pure fungi isolates were obtained using the cutting technique by sub-culturing a previously incubated plate onto a freshly prepared sterile PDA plate, the pure bacterial isolates were obtained using the streaking method by a sub-culturing colony from a previously incubated plate onto a freshly prepared sterile nutrient agar plate. For the isolation and identification of the isolates after incubation, the nutrient agar culture plates with colonies between 30 and 300 were counted and the mean value was expressed as the

colony-forming unit per ml (Cfu/mL). The bacterial isolates were characterized using colonial, morphological and biochemical identification methods. They were further identified using Bergey's manual of determinative bacteriology; the microscopic and macroscopic identification of the fungi isolates was carried out using lactophenol in the cotton blue stain technique.

### 2.2. Statistical Analysis

The data obtained were statistically analyzed using descriptive (mean $\pm$ standard error) and inferential statistics (correlation and analysis of variance (ANOVA)). The means of physicochemical parameters for the sachet water samples subjected to different storage conditions were compared using Duncan's multiple range test and multivariate statistics (principal component analysis (PCA)) were used to express the interrelationship between the physicochemical parameters, microbiological quality and storage conditions (SPSS software version 25 and PAST software version 3).

### 3. Results

#### 3.1. PhysicoChemical Parameters

The highest pH mean concentration was recorded in the sachet water samples used for control ($7.14 \pm 0.24$) and the lowest mean value was observed in the sachet water stored on the plank ($6.68 \pm 0.23$). The lowest water temperature was observed in the sachet water samples stored inside the fridge ($14.90 \pm 1.44$ °C). There is a very high significant different variation ($p > 0.001$) among the pH mean concentrations of the sachet water samples subjected to different storage conditions, as presented in Table 1. The highest mean concentrations of EC, TDS, BOD and calcium ($336.67 \pm 73.69$ µS/cm, $168.33 \pm 34.89$ mg/L, $1.87 \pm 0.39$ mg/L, and $2.97 \pm 0.19$ mg/L) were recorded in the sachet water samples stored on the floor. The lowest DO mean value of $2.01 \pm 0.49$ mg/L was recorded in the water samples stored on the floor. There is a significant difference ($p > 0.05$) among the DO mean concentration of the sachet waters subjected to different storage conditions. There was a significant difference ($p < 0.05$) in the mean concentration of the nitrates, phosphates, and hardness compared with other means of sachet water samples subjected to different storage conditions (Table 1). The Pearson correlation matrix showing the relationship between the physicochemical and microbial quality of the sachet water samples subjected to different storage conditions is presented in Table 2. The pH correlates with the water temperature in all the storage conditions, except in plank, and with the BOD, except in the control sachet water samples. Water temperature correlates with the TBC, TFC, BOD, and DO except in plank storage conditions; EC correlates with the TDS and hardness, except in the fridge storage conditions. The DO clusters with phosphate in all storage conditions, the BOD correlates with TBC, while nitrates clustered with the TBC and phosphates clustered with the TBC (Table 2). The highest mean concentrations of the DO and BOD ($4.50 \pm 0.10$ mg/L and $1.30 \pm 0.10$ mg/L) were observed in the A brand of the sachet water samples compared with the other brands of sachet water among the control water samples, while the pH, water temperature and nitrate and phosphate mean values ($7.54 \pm 0.04$, $26.40 \pm 0.30$ °C, $0.27 \pm 0.02$ mg/L and $0.15 \pm 0.00$ mg/L) were higher in the B brand of the sachet water samples (Table 3). The mean concentrations of the electrical conductivity (EC), TDS, chloride, calcium and hardness ($342.00 \pm 126.00$ µS/cm, $294.00 \pm 101.00$ mg/L, $25.71 \pm 14.53$ mg/L and $2.50 \pm 0.10$ CaCO$_3$ mg/L) were higher in the C brand of sachet water and it is significantly different ($p < 0.05$) among the mean values of the EC, TDS and BOD of the sachet water used in the control water sample. The maximum mean concentrations of the physicochemical parameters such as the pH, water temperature, EC, TDS and chloride ($7.21 \pm 0.11$, $7.20 \pm 0.20$ °C, $361.00 \pm 100.00$ µS/cm, $177.75 \pm 95.12$ mg/L and $21.05 \pm 18.21$ mg/L) were recorded in the C brand of sachet water stored inside the fridge. The highest mean values of the DO, BOD, phosphate and calcium were observed in the B brand sachet water, while the mean values of the nitrate and hardness were recorded in the A brand of sachet water which was subjected to fridge conditions. The highest DO

and nitrate mean concentrations ($2.62 \pm 0.10$ mg/L and $0.19 \pm 0.01$ mg/L) were recorded in the A brand of sachet water stored on the floor, while the water temperature, BOD, phosphate and calcium values were higher in the B brand of sachet water stored on the floor. The highest mean concentration of pH, EC, TDS and chloride was observed in the C brand of sachet water stored on the floor (Table 3). The highest means of the pH, EC, TDS, BOD and calcium ($7.09 \pm 0.09$, $236.00 \pm 130.03$ μS/cm, $132.00 \pm 102.94$ mg/L, $1.35 \pm 0.05$ mg/L and $3.60 \pm 0.10$ mg/L) were recorded in the C brand of sachet water stored on a plank, while the DO and chloride (the B brand of sachet water) and the mean of the water temperature and hardness were higher in the A brand of sachet water which was stored on the plank (Table 3).

**Table 1.** Descriptive statistical analysis of physicochemical parameters and microbial quality of sachet water subjected to different storage conditions.

| Parameters | Control | Storage Conditions | | | Anova | | WHO |
|---|---|---|---|---|---|---|---|
| | | Fridge | Floor | Plank | | | |
| | Mean ± Sem | Mean ± Sem | Mean ± Sem | Mean ± Sem | F | Sig. | |
| pH | $7.14 \pm 0.24$ [a] | $6.77 \pm 0.25$ [a] | $7.07 \pm 0.36$ [a] | $6.68 \pm 0.23$ [a] | 0.657 | 0.601 | 6.5–9.5 |
| Temp. (°C) | $25.48 \pm 0.46$ [b] | $14.90 \pm 1.44$ [a] | $26.34 \pm 0.17$ [b] | $25.40 \pm 0.43$ [b] | 47.401 | 0.000 | 30–35 |
| EC (μS/cm) | $308.47 \pm 28.39$ [a] | $237.67 \pm 63.47$ [a] | $336.67 \pm 73.69$ [a] | $172.83 \pm 34.21$ [a] | 1.896 | 0.209 | 1000 |
| TDS (mg/L) | $237.50 \pm 55.75$ [a] | $117.42 \pm 31.12$ [a] | $168.33 \pm 34.89$ [a] | $91.00 \pm 21.54$ [a] | 2.891 | 0.102 | 500 |
| DO (mg/L) | $4.05 \pm 0.40$ [a] | $2.44 \pm 0.33$ [b] | $2.01 \pm 0.49$ [b] | $2.16 \pm 0.56$ [b] | 4.254 | 0.042 | 7.5 |
| BOD (mg/L) | $1.00 \pm 0.25$ [a] | $1.13 \pm 0.32$ [a] | $1.87 \pm 0.39$ [a] | $0.85 \pm 0.40$ [a] | 1.689 | 0.246 | 7–9 |
| Chloride (mg/L) | $20.92 \pm 3.13$ [a] | $15.94 \pm 3.11$ [a] | $17.93 \pm 4.38$ [a] | $14.65 \pm 3.33$ [a] | 0.599 | 0.633 | 250 |
| Nitrate (mg/L) | $0.14 \pm 0.06$ [a] | $0.09 \pm 0.01$ [a] | $0.21 \pm 0.01$ [b] | $0.17 \pm 0.01$ [b] | 7.752 | 0.010 | 50 |
| Phosphate (mg/L) | $0.09 \pm 0.04$ [a] | $0.04 \pm 0.01$ [b] | $0.10 \pm 0.01$ [a] | $0.06 \pm 0.01$ [ab] | 6.674 | 0.013 | 50 |
| Calcium (mg/L) | $2.67 \pm 0.42$ [a] | $2.37 \pm 0.28$ [a] | $2.97 \pm 0.19$ [a] | $3.32 \pm 0.26$ [a] | 0.839 | 0.218 | 250 |
| Hardness ($CaCO_3$ mg/L) | $2.77 \pm 0.65$ [ab] | $1.33 \pm 0.13$ [a] | $3.33 \pm 0.48$ [b] | $3.27 \pm 0.28$ [ab] | 4.601 | 0.037 | 300 |

Means with a different superscript letter (s) along the same row were significantly different ($p > 0.05$). WHO (2011) Permissible Limits/Standards for Drinking Water. Temp.—temperature, EC—electrical conductivity, TDS—total dissolved solids, DO—dissolved oxygen, BOD—biological oxygen demand.

**Table 2.** Pearson correlation matrix showing the relationship between physicochemical parameters and microbial quality of the sachet water samples subjected to different storage conditions.

Control

| | pH | Temp. | EC | TDS | DO | BOD | $NO_3^-$ | $PO_4^{3-}$ | $Ca^{2+}$ | Hardness | TFC | TBC |
|---|---|---|---|---|---|---|---|---|---|---|---|---|
| pH | 0.00 | | | | | | | | | | | |
| Temp. | 0.85 *** | 0.00 | | | | | | | | | | |
| EC | −0.90 | −0.54 | 0.00 | | | | | | | | | |
| TDS | 0.95 *** | 0.97 *** | −0.72 | 0.00 | | | | | | | | |
| DO | −0.15 | 0.99 *** | 0.56 * | 0.17 | 0.00 | | | | | | | |
| BOD | −0.25 | −0.73 | −0.19 | −0.54 | −0.92 | 0.00 | | | | | | |
| Nitrate | 0.71 ** | 0.98 *** | −0.34 | 0.89 *** | 0.59 * | −0.86 | 0.00 | | | | | |
| Phosphate | 0.13 | 0.63 ** | 0.31 | 0.43 | 0.96 *** | −0.99 | 0.79 ** | 0.00 | | | | |
| $Ca^{2+}$ | −0.56 | −0.91 | 0.15 | −0.79 | −0.74 | 0.94 *** | −0.98 | −0.89 | 0.00 | | | |
| Hardness | −0.99 | −0.77 | 0.95 *** | −0.90 | 0.28 | 0.13 | −0.61 | 0.00 | 0.45 | 0.00 | | |
| TFC | −0.93 | −0.59 | 0.99 *** | −0.77 | 0.50 * | −0.12 | −0.40 | 0.25 | 0.21 | 0.97 *** | 0.00 | |
| TBC | −0.15 | 0.40 | 0.56 * | 0.17 | 1.00 *** | −0.92 | 0.59 * | 0.96 *** | −0.74 | 0.27 | 0.50 * | 0.00 |

**Table 2.** *Cont.*

| Sachet water stored inside the fridge | | | | | | | | | | | |
|---|---|---|---|---|---|---|---|---|---|---|---|
| pH | 0.00 | | | | | | | | | | |
| Temp. | 0.99 *** | 0.00 | | | | | | | | | |
| EC | −0.70 | −0.58 | 0.00 | | | | | | | | |
| TDS | −0.69 | −0.57 | 0.99 *** | 0.00 | | | | | | | |
| DO | −0.04 | 0.81 *** | 0.75 ** | 0.75 ** | 0.00 | | | | | | |
| BOD | 0.85 *** | 0.92 *** | −0.21 | −0.20 | 0.49 | 0.00 | | | | | |
| Nitrate | −0.95 | −0.99 | 0.44 | 0.43 | −0.27 | −0.97 | 0.00 | | | | |
| Phosphate | −0.03 | 0.12 | 0.74 ** | 0.75 ** | 0.99 *** | 0.50 * | −0.28 | 0.00 | | | |
| Ca²⁺ | 0.79 ** | 0.88 *** | −0.11 | −0.10 | 0.58 * | 0.99 *** | −0.94 | 0.59 * | 0.00 | | |
| Hardness | −0.85 | −0.92 | 0.21 | 0.20 | −0.49 | −1.00 | 0.97 *** | −0.50 | −0.99 | 0.00 | |
| TFC | 0.98 *** | 0.99 *** | −0.52 | −0.51 | 0.18 | 0.95 *** | −1.00 | 0.19 | 0.91 *** | −0.94 | 0.00 |
| TBC | 0.99 *** | 0.98 *** | −0.73 | −0.72 | −0.09 | 0.82 *** | −0.93 | −0.08 | 0.76 ** | −0.82 | 0.96 *** | 0.00 |

| Sachet water stored on the floor | | | | | | | | | | | |
|---|---|---|---|---|---|---|---|---|---|---|---|
| pH | 0.00 | | | | | | | | | | |
| Temp. | 0.52 * | 0.00 | | | | | | | | | |
| EC | −0.54 | 0.44 | 0.00 | | | | | | | | |
| TDS | −0.55 | 0.43 | 0.99 *** | 0.00 | | | | | | | |
| DO | −0.26 | 0.69 ** | 0.95 *** | 0.95 *** | 0.00 | | | | | | |
| BOD | 0.92 *** | 0.93 *** | −0.83 | −0.84 | −0.63 | 0.00 | | | | | |
| Nitrate | −0.92 | 0.81 *** | 0.16 | 0.17 | −0.14 | −0.68 | 0.00 | | | | |
| Phosphate | 0.11 | 0.91 *** | 0.78 ** | 0.77 ** | 0.93 *** | −0.31 | −0.49 | 0.00 | | | |
| Ca²⁺ | 0.33 | 0.98 *** | 0.62 ** | 0.61 ** | 0.82 *** | −0.08 | 0.78 ** | 0.48 | 0.00 | | |
| Hardness | −0.92 | −0.15 | 0.82 *** | 0.83 *** | 0.61 ** | −1.00 | 0.69 | 0.29 | 0.06 | 0.00 | |
| TFC | −0.04 | −0.88 | −0.82 | −0.81 | −0.95 | 0.37 | 0.43 | −1.00 | −0.96 | −0.35 | 0.00 |
| TBC | 0.95 *** | 0.24 | −0.77 | −0.78 | −0.54 | 0.99 *** | −0.76 | 0.97 *** | 0.03 | −1.00 | 0.26 | 0.00 |

| Sachet water stored on the plank | | | | | | | | | | | |
|---|---|---|---|---|---|---|---|---|---|---|---|
| pH | 0.00 | | | | | | | | | | |
| Temp. | −0.83 | 0.00 | | | | | | | | | |
| EC | −0.57 | 0.02 | 0.00 | | | | | | | | |
| TDS | −0.57 | 0.02 | 1.00 *** | 0.00 | | | | | | | |
| DO | 0.90 *** | −0.99 | −0.16 | −0.16 | 0.00 | | | | | | |
| BOD | 0.99 *** | 0.73 ** | −0.70 | −0.70 | 0.82 *** | 0.00 | | | | | |
| Nitrate | 0.74 ** | −0.24 | −0.98 | −0.97 | 0.37 | 0.84 *** | 0.00 | | | | |
| Phosphate | −0.88 | 0.47 | 0.89 *** | 0.89 *** | −0.59 | −0.94 | −0.97 | 0.00 | | | |
| Ca²⁺ | 0.88 *** | −1.00 | −0.11 | −0.11 | 0.99 *** | 0.79 ** | 0.33 | −0.55 | 0.00 | | |
| Hardness | −0.92 | 0.56 * | 0.84 *** | 0.84 *** | −0.67 | −0.97 | −0.94 | 0.99 *** | −0.63 | 0.00 | |
| TFC | −0.57 | 0.93 *** | −0.34 | −0.34 | −0.87 | −0.43 | 0.13 | 0.11 | −0.89 | 0.21 | 0.00 |
| TBC | 0.98 *** | −0.93 | −0.38 | −0.38 | 0.97 *** | 0.93 *** | 0.58 * | −0.76 | 0.96 *** | −0.82 | −0.74 | 0.00 |

\* (Significant different $p < 0.05$) \*\* (Significant different $p < 0.01$) and \*\*\* (Significant different $p < 0.001$).

**Table 3.** Descriptive statistical analysis of physicochemical parameters and microbial quality of the three sachet water brands subjected to different storage conditions.

| | Control | | | | |
|---|---|---|---|---|---|
| | **Three Brand of Sachet Water Samples** | | | | |
| **Parameters** | **A** | **B** | **C** | ***p*-Value** | **WHO** |
| | **Mean ± Sem** | **Mean ± Sem** | **Mean ± Sem** | | |
| pH | 6.71 ± 0.03 | 7.54 ± 0.04 | 7.18 ± 0.08 | 0.075 | 6.5–9.5 |
| Temp. (°C) | 24.98 ± 0.13 | 26.40 ± 0.30 | 25.05 ± 0.05 | 0.188 | 30–35 |
| EC (µS/cm) | 252.00 ± 122.00 | 331.40 ± 130.04 | 342.00 ± 126.00 | 0.006 | 1000 |
| TDS (mg/L) | 126.00 ± 70.10 | 292.50 ± 125.00 | 294.00 ± 101.00 | 0.03 | 500 |
| DO (mg/L) | 4.50 ± 0.10 | 4.40 ± 0.10 | 3.25 ± 0.05 | 0.247 | 7.5 |
| BOD (mg/L) | 1.30 ± 0.10 | 1.20 ± 0.20 | 0.50 ± 0.10 | 0.003 | 7–9 |

**Table 3.** *Cont.*

| Parameters | Control Three Brand of Sachet Water Samples | | | *p*-Value | WHO |
|---|---|---|---|---|---|
| | **A** | **B** | **C** | | |
| | **Mean ± Sem** | **Mean ± Sem** | **Mean ± Sem** | | |
| Chloride (mg/L) | 15.03 ± 8.51 | 22.01 ± 10.41 | 25.71 ± 14.53 | 0.944 | 250 |
| Nitrate (mg/L) | 0.10 ± 0.00 | 0.27 ± 0.02 | 0.06 ± 0.00 | 0.442 | 50 |
| Phosphate (mg/L) | 0.12 ± 0.00 | 0.15 ± 0.00 | 0.01 ± 0.00 | 0.334 | 50 |
| Calcium (mg/L) | 2.25 ± 0.15 | 2.25 ± 0.00 | 3.50 ± 0.10 | 0.611 | 250 |
| Hardness (CaCO$_3$ mg/L) | 4.00 ± 0.10 | 1.80 ± 0.10 | 2.50 ± 0.10 | 0.774 | 300 |
| **Fridge** | A | B | C | *p*-Value | WHO |
| Parameters | Mean ± Sem | Mean ± Sem | Mean ± Sem | | |
| pH | 6.34 ± 0.04 | 6.75 ± 0.04 | 7.21 ± 0.11 | 0.673 | 6.5–9.5 |
| Temp. (°C) | 12.25 ± 0.25 | 15.25 ± 0.25 | 17.20 ± 0.20 | 0.548 | 30–35 |
| EC (µS/cm) | 150.00 ± 110.00 | 202.00 ± 105.03 | 361.00 ± 100.00 | 0.049 | 1000 |
| TDS (mg/L) | 74.00 ± 50.02 | 100.50 ± 53.09 | 177.75 ± 95.12 | 0.000 | 500 |
| DO (mg/L) | 2.11 ± 1.10 | 3.10 ± 2.01 | 2.10 ± 1.93 | 0.219 | 7.5 |
| BOD (mg/L) | 1.40 ± 0.20 | 1.50 ± 0.05 | 0.50 ± 0.20 | 0.538 | 7–9 |
| Chloride (mg/L) | 10.31 ± 3.04 | 16.45 ± 7.85 | 21.05 ± 18.21 | 0.144 | 250 |
| Nitrate (mg/L) | 0.13 ± 0.02 | 0.10 ± 0.00 | 0.09 ± 0.00 | 0.467 | 50 |
| Phosphate (mg/L) | 0.04 ± 0.00 | 0.05 ± 0.01 | 0.04 ± 0.00 | 0.326 | 50 |
| Calcium (mg/L) | 1.80 ± 0.10 | 2.70 ± 0.10 | 2.60 ± 0.10 | 0.426 | 250 |
| Hardness (CaCO$_3$ mg/L) | 1.60 ± 0.10 | 1.20 ± 0.10 | 1.20 ± 0.10 | 0.462 | 300 |
| **Floor** | A | B | C | *p*-Value | WHO |
| Parameters | Mean ± Sem | Mean ± Sem | Mean ± Sem | | |
| pH | 6.40 ± 0.04 | 7.15 ± 0.16 | 7.65 ± 0.15 | 0.761 | 6.5–9.5 |
| Temp. (°C) | 26.08 ± 0.65 | 26.65 ± 0.15 | 26.30 ± 0.30 | 0.656 | 30–35 |
| EC (µS/cm) | 220.00 ± 100.00 | 317.00 ± 271.00 | 473.00 ± 100.01 | **0.031** | 1000 |
| TDS (mg/L) | 112.50 ± 87.50 | 160.00 ± 86.00 | 232.50 ± 150.31 | 0.205 | 500 |
| DO (mg/L) | 2.62 ± 0.10 | 1.02 ± 0.18 | 2.38 ± 0.16 | 0.579 | 7.5 |
| BOD (mg/L) | 1.10 ± 0.10 | 2.40 ± 0.10 | 2.10 ± 0.10 | 0.792 | 7–9 |
| Chloride (mg/L) | 9.74 ± 4.01 | 19.34 ± 13.21 | 24.71 ± 14.01 | 0.449 | 250 |
| Nitrate (mg/L) | 0.19 ± 0.01 | 0.16 ± 0.01 | 0.16 ± 0.01 | 0.977 | 50 |
| Phosphate (mg/L) | 0.02 ± 0.01 | 0.10 ± 0.02 | 0.06 ± 0.00 | 0.822 | 50 |
| Calcium (mg/L) | 2.70 ± 0.10 | 3.35 ± 0.15 | 2.85 ± 0.05 | 0.751 | 250 |
| Hardness (CaCO$_3$ mg/L) | 4.00 ± 0.10 | 3.60 ± 0.10 | 2.40 ± 0.10 | 0.478 | 300 |
| **Plank** | A | B | C | *p*-Value | WHO |
| Parameters | Mean ± Sem | Mean ± Sem | Mean ± Sem | | |
| pH | 6.29 ± 0.04 | 6.67 ± 0.02 | 7.09 ± 0.09 | 0.859 | 6.5–9.5 |
| Temp. (°C) | 26.25 ± 0.25 | 24.95 ± 0.15 | 25.00 ± 0.10 | 0.399 | 30–35 |
| EC (µS/cm) | 118.50 ± 93.50 | 164.00 ± 74.00 | 236.00 ± 130.03 | 0.328 | 1000 |
| TDS (mg/L) | 59.00 ± 23.09 | 82.00 ± 24.00 | 132.00 ± 102.94 | 0.304 | 500 |
| DO (mg/L) | 1.05 ± 0.10 | 2.62 ± 1.80 | 2.82 ± 2.01 | 0.886 | 7.5 |
| BOD (mg/L) | 0.05 ± 0.05 | 1.15 ± 0.05 | 1.35 ± 0.05 | 0.876 | 7–9 |
| Chloride (mg/L) | 8.02 ± 3.28 | 18.42 ± 13.09 | 17.51 ± 13.78 | 0.959 | 250 |
| Nitrate (mg/L) | 0.16 ± 0.01 | 0.15 ± 0.01 | 0.19 ± 0.01 | 0.468 | 50 |
| Phosphate (mg/L) | 0.07 ± 0.00 | 0.07 ± 0.01 | 0.03 ± 0.01 | 0.878 | 50 |
| Calcium (mg/L) | 2.80 ± 0.10 | 3.55 ± 0.05 | 3.60 ± 0.10 | 0.861 | 250 |
| Hardness (CaCO$_3$ mg/L) | 3.60 ± 0.10 | 3.50 ± 0.10 | 2.70 ± 0.10 | 0.865 | 300 |

### 3.2. Microbial Quality

A total number of 15 fungal species and 4 bacteria species were identified from the three selected brands of sachet water examined during the study period belonging to 8 classes, namely: Eurotiomycete (8), Saccharomycete (1), Tremellomycete (1),

Malasseziomycete (1), Actinomycetia (1), Sordariomycete (2), Ascomycete (1) and Gammaproteobateria (4), as shown in Table 4. *Aspergillus niger* and *Penicillium* spp. record the highest species occurrence of 25% among the fungus identified, while *Proteus* sp. record the highest species occurrence of 50% among the bacteria isolated from this study. The highest total bacterial counts were recorded in the brand of sachet water stored on the floor, followed by the plank and fridge, while the total fungal counts were higher in the sachet water stored on the plank, followed by the floor and fridge (Figure 1). The highest total fungal counts were observed in the A and C brands of sachet water samples stored on the plank, while the total bacteria counts were recorded in the C brand of sachet water samples stored on the floor and plank, as presented in Figure 2. Principal component analysis showed a positive strong correlation between the pH, TDS, DO, BOD, phosphate, hardness, temperature, nitrate, electrical conductivity (EC), calcium and total bacteria count and the sachet water stored on the floor (positive *x*-axis) than with the sachet water stored on the plank (positive *y*-axis), while there is a negative relationship between the total fungal counts with the sachet water stored inside the fridge (Figure 3).

**Table 4.** The occurrence of fungal and bacterial species identified from three brands of sachet water samples subjected to different storage conditions.

| Class | Microbial Species | Control | | | Fridge | | | Floor | | | Plank | | | % Occurrence |
|---|---|---|---|---|---|---|---|---|---|---|---|---|---|---|
| | | A | B | C | A | B | C | A | B | C | A | B | C | |
| | | | | | | | | **Fungi** | | | | | | |
| Eurotiomycete | *Aspergillus fumigatus* | | | | | | | + | + | | | | | 16.7 |
| | *Aspergillus nidulaus* | | | | | | | + | | | | | | 8.3 |
| | *Aspergillus niger* | + | | | | | | + | | | + | | | 25.0 |
| | *Exophiala jeanseimei* | | | | | | | | | + | | | | 8.3 |
| | *Fonsecaea pedrosoi* | | | | | + | | | | + | + | | | 25.0 |
| | *Malbranchea cinnamonea* | | + | | | | | | | | | | | 8.3 |
| | *Penicillium* spp. | | | + | + | | | + | | | | | | 25.0 |
| | *Trichophyton mentagraphytes* | | | | | + | | | | | | | | 8.3 |
| Malasseeziomycete | *Malassezia pachydermatis* | | + | | | | | | | | | | | 8.3 |
| Actinomycetia | *Nocardiopsis dossonvillei* | + | | | | | | | | | | | | 8.3 |
| Sordariomycete | *Pseudallescheria boydii* | | | | | | | | + | | | | | 8.3 |
| | *Scopulariopsis* spp. | | | | | | | | | | | | + | 8.3 |
| Ascomycetes | *Wangiella dermatitidis* | | | | | | + | | | | | | | 8.3 |
| Saccharomycete | *Candida krusei* | | | | | | | | | + | | + | | 16.7 |
| Tremellomycete | *Cryptococcus neoformans* | | | | | | | | | + | | | | 8.3 |
| | | | | | | | | **Bacteria** | | | | | | |
| Gammaprotebacteria | *Aeromonas* sp. | | + | | | | | + | + | | | | | 25.0 |
| | *Proteus* sp. | + | | | | + | + | | | + | | + | + | 50 |
| | *Vibro* sp. | | | | | + | + | | | | | | | 16.7 |
| | *Yersinia* sp. | | | | | | | + | | + | | | + | 25.0 |
| | Total microbial counts | 3 | 3 | 2 | 2 | 3 | 2 | 5 | 3 | 6 | 3 | 3 | 3 | |

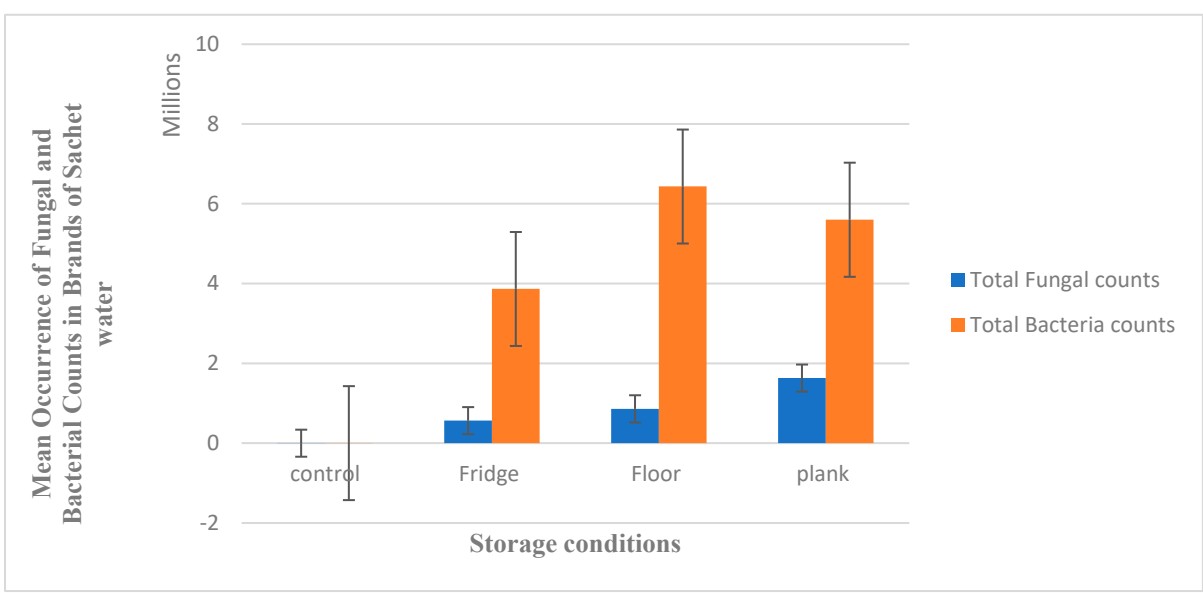

**Figure 1.** Bar chart diagram showing the occurrence of the total fungal and bacterial counts in the different storage conditions.

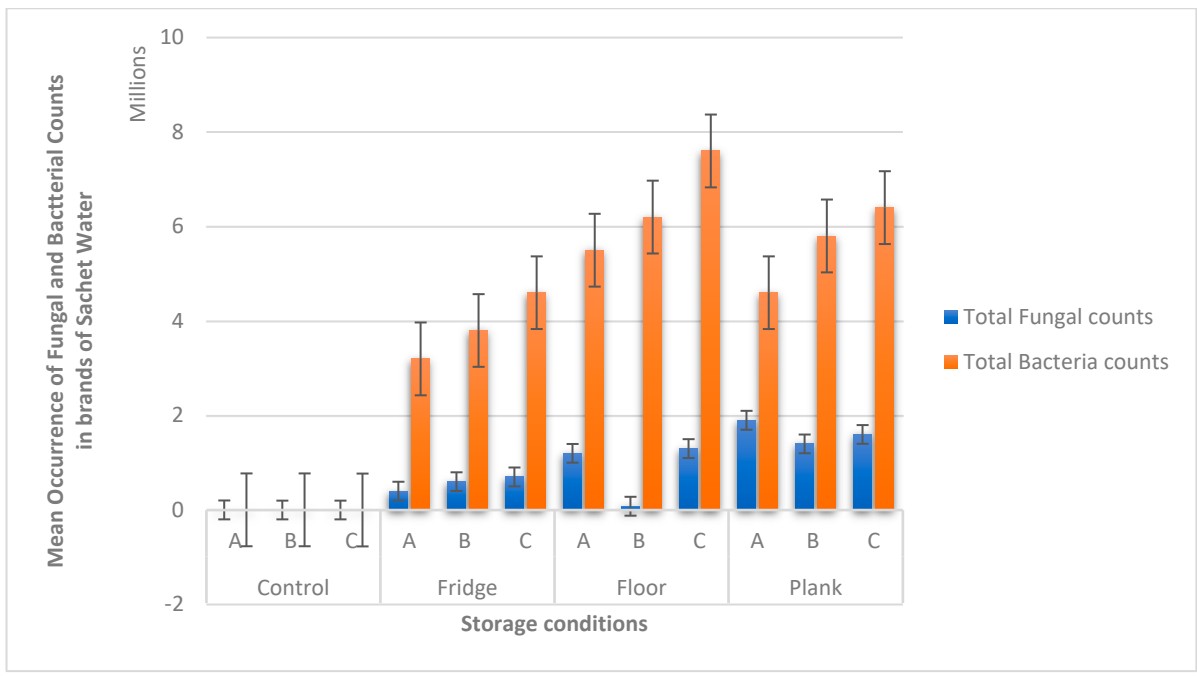

**Figure 2.** Bar chart diagram showing the occurrence of the total fungal and bacterial counts in the three selected brands of sachet water subjected to different storage conditions.

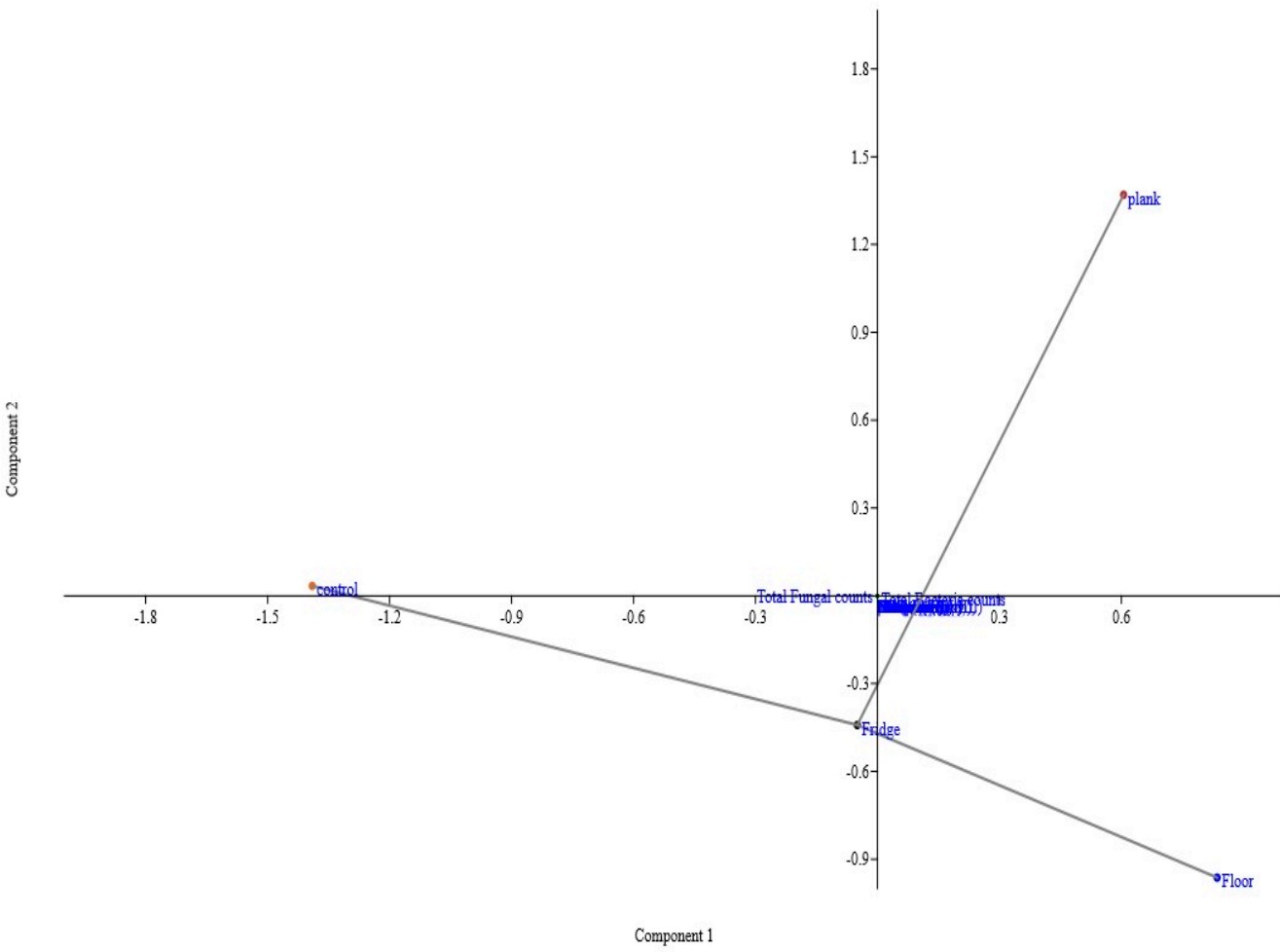

**Figure 3.** PCA showing the relationship between different storage conditions, physicochemical and microbial quality of the selected brands of sachet water.

## 4. Discussion

Most of the physicochemical parameters examined for the three selected brands of sachet water subjected to different storage conditions were within [12,16] the permissible limits for drinking water, except for the pH of the A brand of sachet water that fell within the slightly acidic range. This could be attributed to the high temperature, which has a direct link with pH; the lower the pH of water, the higher the temperature. This instead could be due to the production of basic metabolic waste which encourages the regrowth of the bacteria population [29]. There are decreases in the pH concentrations of the sachet water samples subjected to different storage conditions compared with the control sachet water samples. This contradicts the work of [10] who reported that the rapid development of bacteria in sachet water is aided by the availability of organic materials, nutrients, dissolved oxygen present in the water, and the prolonged storage of water. The pH range values recorded during this study were similar to the study conducted on sachet water quality by [7] in Sierra Leone, ref. [30] in Kazaure metropolis, Jigawa state [31], and Ghana. The pH control is essential at every stage of the water treatment and during the storage period to ensure a satisfactory water clarification, and its disinfection to prevent microbial regrowth [32]. The decrease in the DO values obtained in all three storage conditions aligned with the report by [33]. The decreasing trend obtained in the DO mean value across all storage conditions could be a result of the polythene bags used in the sachet water package which are made of synthetic petroleum that reacts with hot temperatures; this can lead to water quality deterioration because it is weather susceptible. When the sun's rays or heat melts some of the synthetic petroleum in the sachet, it is mixed with the

water and this can be carcinogenic to the consumers. A high biological oxygen demand (BOD) mean was recorded in all the sachet water subjected to different storage conditions, which aligns with the work of [33] who reported the highest BOD values of 1.8 mg/L in sachet water sold in Zaria. The high biological oxygen demand (BOD) mean values observed during this study indicate high microbial activities that lead to a reduction in oxygen concentration. This correlation showed that there is a relationship between water temperature and BOD, which indicate that temperature played a major role in water quality analysis since it affects both the physicochemical and bacteriological processes, such as the absorption of chemicals and microbial growth. High water temperature mean values were observed in sachet water samples stored on the floor which could be due to the reaction between the heat generated by the floor, influencing the sachet water temperature. Similar water temperature ranges were recorded in the analysis of sachet water quality by [7] in Sierra Leone; [34,35] in Dutse Metropolis, Jigawa State; and Irele, Ondo State [36]. There will be a great health risk challenge for consumers drinking these sachet waters because the cellophane bag used for its packaging contains large doses of dioxin (bisphenol A (BPA), phthalate, and other chemicals of concern). These chemicals are harmful to health when exposed to sunlight with a temperature > 28 °C as they leach antimony trioxide, which reacts with water. Moreover, a variety of additional organic components and chemical pollutants that may affect the flavor of the water may also be affected by temperature [9]. These chemicals can be a carcinogen, hazardous, and can damage the immune system of the consumer, and can also lead to the failure of the kidney and liver of the water consumers. High water temperatures encourage the growth of microbes, which may worsen issues with the taste, odor, and color of the water [5].

The low conductivity and TDS concentrations observed during this study could be due to the presence of low ions as a result of different treatment processes undergone by the sachet water, or the cool nature of the water. These findings are similar to the report of [37], but they contradict the work of [36]. The conductivity values are influenced by temperature; the cooler the water the lower the conductivity, and vice versa. The low total dissolved solid (TDS) could be a result of appropriate membrane filtration procedures used in the water purification process to filter both organic and inorganic contaminants. The mean values of the chemical parameters such as the chloride, nitrate, phosphate, calcium and hardness were within [12,16] the stipulated standards limit for drinking water and fell within the range obtained by [38] in Sango-ota, [39] but contradicts the report of [35] in Dutse Metropolis; [36] in Irele; and [40] in Ghana. The high concentration of the hardness recorded in the sachet water samples stored on the floor might be due to prolonged storage which can activate the growth of algal and other biochemical activities. However, the low concentrations of calcium and magnesium recorded in this study fell under the soft water category which has a great tendency of causing health problems for consumers. They could result in a high risk of fracture in children (calcium is essential for the nervous system and for bone formation) and may increase morbidity and mortality in patients with cardiovascular diseases (CVDs) compared to hard water [41,42]. The low mean values of nitrate and phosphate in this study conform with the report of [43]. The low concentration of limiting nutrients like nitrates and phosphates in the sachet water samples could be limiting factors for microbial regrowth in water, even when the environmental factors are favorable [44]. Principal component analysis (PCA) does not exclude any samples or attributes from its analysis (variables). Instead, it reduces the overwhelming number of dimensions by constructing principal components (PCs). A 90° angle implies no association between two characteristics, whereas a small angle suggests a positive correlation. Loading charts also provide a suggestion as to how variables correlate with one another. This is a positive strong correlation between all physicochemical parameters and the microbial quality of the sachet water quality with floor and plank storage conditions, except for the total fungal count that correlates with the fridge storage condition; this shows that the period of storage and types storage conditions have a great impact on sachet water quality.

A total number of 19 microbial species were identified from the three selected brands of sachet water sold within the Samaru community, namely: *Aspergillus niger*, *A. nidulans*, *A. fumigatus*, *Candida krusei*, *Cryptococcus neoformas*, *Exophiala jeanseimei*, *Fonsecaea pedrosoi*, *Malassezia pachydermatis*, *Malbranchea cinnamomea*, *Nocardiopsis dossonvillei*, *Penicillium* spp., *Pseudallescheria boydii*, *Scopulariopsis* spp., *Trichophyton mentagraphyte*, *Wangiella (Exophiala) dermatitidis*, *Aeromonas* sp., *Proteus* sp., *Vibro* sp. and *Yersinia* sp. The high microbial species occurrence that was recorded during this study could be attributed to the availability of oxygen in all the brands of sachet water samples analyzed and favorable temperatures that encourage microbial growth during storage, as well as the trace amounts of nutrients arising from the packaging materials [44]. The result of this study contradicts the report by [45] that observed a low percentage of fungal occurrence in six brands of sachet water marketed in Zaria. The total aerobic heterotrophic bacteria can grow to levels that could be dangerous to humans when packaged water is stored for an extended period of time in a favorable environment [46]. Although the microbial quantity levels in processed water are often initially low, they can evolve rapidly to high levels based on the method of storage. The results obtained from this study contradict the report by [47] that found that the decline in the bacterial population can be attributed to the death of the resident bacteria during the storage period due to the depletion of nutrients. The highest microbial species occurrence was recorded in the brand C sachet water sample stored on the floor with six species, followed by the brand A sachet water sample stored on the floor with five species. The highest total fungi counts were observed in the brand C sachet water sample stored on the floor, while the highest bacterial counts were recorded in the brand A sachet water sample stored on the plank. The rapid growth nature of the bacteria can increase the total aerobic heterotrophic bacteria in these water samples after the sachet water is packaged and stored at room temperature for a long period [10].

The total bacterial count (TBC) was higher in the sachet water stored on the floor while the total fungal count (TFC) was higher in the sachet water stored on the plank, compared with the low total bacterial count and total fungal count recorded in fridge storage conditions. The floor storage condition favors the regrowth of microbial species due to an alteration in the sachet water quality from a prolonged storage, while only microorganisms which can adapt to low-temperature conditions grow in fridge storage. Most of the microbial species isolated from the selected brands of sachet water during this study are tolerant to stressful environmental conditions, such as an extremely low pH, nitrogen and calcium limitation, high temperatures and radiation resistance. They are cosmopolite organisms and well known to be pathogenic, particularly *Aspergillus*, *Candida*, *Nocardiopsis*, *Aeromonas*, *Vibro* and *Yersinia* are health risks [48] and cause taste and odor in water [49]. They cause all forms of diseases, including allergies to fungal antigens, strong allergenic skin irritants or may cause infections in immune-suppressed individuals with acquired immune deficiency syndrome, in cancer patients, organ transplant patients, persons with asthma or various respiratory problems, as well as in the production of toxins. It could also cause a direct invasion of hosts; it is capable of infecting healthy hosts and causing diseases ranging from mucosal to life-threatening disseminated infections [50,51]. The fungus *Aspergillus* is a causative agent of kidney and liver disorders, allergic sinusitis burns, otitis media and is capable of increasing the risk of invasive infections. Additionally, *Scopulariopsis* spp. causes the development of fungus balls in preformed pulmonary cavities, keratitis, posttraumatic endophthalmitis, disseminated skin lesions in AIDS patients, granulomatous subcutaneous infection, invasive hyalohyphomycosis, pneumonia in leukemic patients, endocarditis related to valvuloplasty or prosthetic valves and fatal disseminated infection following bone marrow transplantation are among the severe human mycoses. Similarly, *Pseudallescheria boydii* can cause the death of a patient who is immunocompromised and in near-drowning pneumonia patients. *Fonsecaea pedrosoi* cause human chromoblastomycosis, a chronic fungal infection localized to the skin and subcutaneous tissue. *Exophiala jeanselmei* has versatile adaptability and acts as an opportunistic pathogen. *Cryptococcus neoformans* causes fungal meningitis and encephalitis, especially as a secondary infection for AIDS

patients, and often causes lung disease, making it a particularly dangerous fungus. Trichophyton mentagrophytes cause zoonotic stain disease. *Nocardiopsis dassonvelli* can cause severe, suppurative pulmonary infection, while *Wangiella dermatitidis* causes mycosis in humans called Phaeohypomycosis, e.g., a fever or a new heart murmur. Kelley [52] concluded that mycotoxins and other metabolites can be produced by fungi in water and can be extremely diluted in water, thus they are perhaps of minor concern. Nevertheless, water stored for a prolonged period of time may cause an increase in mycotoxin concentrations.

The study revealed that many of the taxa isolated have the potential to secrete pigment (melanin) which provides protection against a range of stresses, thus making them to developed resistance to survive the water treatment process. Fungi and bacteria can also regrow in poorly treated package water depending on the methods of storage and when the environmental condition favors their growth [53]. Based on the report of [52] sand filtration was better for the removal of fungi than coagulation with iron. Chlorination used for the disinfection of water was found to be insufficient against microorganisms in water [54], whereas the use of chlorine dioxide and ozone was reported as the most effective water treatment methods against fungal spores [38]. A recently adopted water treatment method in developed countries is ultraviolet radiation and, with respect to fungi in drinking water, ultraviolet sensitivity to fungi is often related to pigmentation. However, fungi with pigmented spores, such as *Aspergillus* and *Penicillium*, have better protection against radiation and are less sensitive to ultraviolet light [32].

## 5. Conclusions

Based on the findings of this study, we showed that sachet water stored on the floor has the highest fungal and bacteria species occurrence compared to other storage conditions. All the physicochemical parameters examined for all of the sachet water exposed to different storage conditions were within the WHO and NSDWQ permissible limits, but low DO values were observed in sachet water stored on the floor and high BOD values were recorded in all storage conditions. We also showed that storage conditions and periods of time can alter the physicochemical quality and encourage the regrowth of microorganisms in water under a favorable environment after water treatment. All the brands of sachet water subjected to different conditions were contaminated with pathogens that are of health risk for human consumption. Therefore, it is essential to examine sachet water periodically to determine the degree of deterioration with a view to deciding whether it meets the quality requirement of potable water. There should be a monitoring program to ensure strict adherence to local and international standards for potable drinking water to be free of microbial indicators of fecal contamination, and the coliform count per 100 mL should be zero for human consumption.

It is essential to carry out further studies to determine how an increase in temperature affects the quality of packaged water and its health risk to consumers since Packaged water is packed in plastic or polythene (nylon) which can react with water when exposed to sunlight with a temperature > 28 °C because the temperature varies in different regions.

**Author Contributions:** T.A.A. and M.A.; writing—original draft preparation, A.T.O. and A.A.A.; writing—review and editing, U.B., N.A.M. and I.A.A.; visualization, A.C.B. All authors have read and agreed to the published version of the manuscript.

**Funding:** This research received no external funding.

**Data Availability Statement:** Not applicable.

**Acknowledgments:** Our regards go to the Staff of the Department of Microbiology and Veterinary Medicine, Ahmadu Bello University, Zaria, for their assistants during the Lab. Analysis.

**Conflicts of Interest:** The authors declare no conflict of interest.

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
