# Peer review of "Effects of Prolonged Storage Condition on the Physicochemical and Microbiological Quality of Sachet Water and Its Health Implications: A Case Study of Selected Water Brands Sold within Samaru Community, Northwest Nigeria"

_2036-7481, doi:10.3390/microbiolres13040051_

Round 1

Reviewer 1 Report (Previous Reviewer 1)

the authors modified and improved the quality of the article and as a result I recommended its publication.

Author Response

Thank you for criticizing this work

Reviewer 2 Report (New Reviewer)

Dear Editor and Authors

The manuscript presents an interesting and necessary proposal, in addition to being well written. However, the title has an inadequacy with regard to "risk assessment". In fact, confusion over the term is common.

A health risk assessment is a systematic assessment consisting of four steps: 1) Hazard Identification; 2) Dose-Response Assessment; 3) Exposure Assessment and 4) Risk Characterization. However, the manuscript presents results related to the "Exposure Assessment" step, presenting quantitative and qualitative results. Therefore, I suggest revising the term "risk assessment" in the title to a more appropriate term. I consider that the manuscript can be published in Microbiology Research after a minor review.

Minor comments:

Line 27: it is necessary to write in full in the first mention of WHO and NSDWQ.

Line 103: it is necessary to write “five” in full

Author Response

Reviewer comments

  1. The manuscript presents an interesting and necessary proposal, in addition to being well written. However, the title has an inadequacy with regard to "risk assessment". In fact, confusion over the term is common.

Responses

The title of the manuscript has been changed to “The effect of prolonged storage conditions on fungal, bacteriological, Physico-chemical parameters and its health implication: A case study of selected sachet water brands sold within Samaru community, Northwest Nigeria”

  1. Line 27: it is necessary to write in full in the first mention of WHO and NSDWQ.

Responses

WHO and NSDWQ has been written in full “World Health Organization and Nigeria Standard for Drinking Water Quality”. Check line 32

  1. Line 103: it is necessary to write “five” in full

Revision

Reviewer 3 Report (New Reviewer)

This article investigates the quality deterioration in sachet water. The writing is clear. However, the impacts on wider readership need to be elaborated. For scientific purposes, it will be good to link to current research gaps

Author Response

Response

It is essential to carry out further studies to determine how the increase in temperature affects the quality of packaged water and its health risk to consumers. Since Packaged water is packed in plastic or polythene (nylon) which can react with water when exposed to sunlight with a temperature > 28 °C because the temperature varies in different regions.

Check line 426 to 429

This manuscript is a resubmission of an earlier submission. The following is a list of the peer review reports and author responses from that submission.

Round 1

Reviewer 1 Report

2. The Introduction section is, in my opinion, too long; that's why I would recommend that this section be kept within the limits of the topic addressed.

3. The same observation in the case of the Materials and Methods section; the authors should retain the essential elements that ensure the reproducibility of the experiments

4. I also recommend the authors to check the way of writing the references (eg, APHA et al (2005) - line 23).

Author Response

Reviewer 1

  1. The work submitted for evaluation provides information on the physico-chemical and microbiological composition of sachet waters consumed in a selected area of Nigeria. Given the problem of access to water in Africa, the idea of the thesis is valid. However, the form of the study is at a very poor level. This applies to both the technical and content layers. The methodological approach is presented in a very unclear manner. I do not know why water was considered to be stored in different places (apart from the fridge, of course). 

(a). Given the problem of access to water in Africa, the idea of the thesis is valid.

Response:  Thank you for your thorough review and salient observations.

Revised, check lines 50 – 52

(b). However, the form of the study is at a very poor level. This applies to both the technical and content layers.

Response:  it has been rewritten

(c). The methodological approach is presented in a very unclear manner.

Response: It has been revised

(d). I do not know why water was considered to be stored in different places (apart from the fridge, of   course). 

Response: In Africa, most manufacturers when they produce sachet water, they stored it on ordinary floor or plank, likewise the distributors, retailers and vendors while some consumers store it inside fridge. This is the main reason for carry-out this research to investigate the impact of different storage and prolong storage condition on sachet water quality

  1. Does not present the most important achievements of the work, but contains unexplained abbreviations and methodological elements. The authors need to consider in which mode they are writing: indicative or presumptive? The passage needs to be re-edited.

Response: It has been revised

  1. Keywords must not repeat elements of the title

Response: the repeated elements in keywords has been deleted and replaces with other words

  1. Passages on lines 77-79 indicate that the research presented is not new and is secondary to existing studies. Lines 82-83 contain content that has already been presented above in the body of the study. The main objective should be the verification of the hypothesis, obviously based on the information gained. This chapter needs to be rethought and rewritten.

Response: The research is a novel report for the study area to the best of the knowledge of the researchers/authors. The introductory chapter has been reconciled in order to nullify the suggested tautology as pointer out. The wordings of the chapter have also been reduced and the grammatical errors have been properly corrected.

  1. Material and methods. I do not understand the importance of describing the study area in the form of an independent point. What is the significance of providing information about the area. In addition, the inclusion of a third-level subsection "sampling collection" unnecessarily complicates the structure of the work. The description of the procedure for the collection of empirical data contains, besides important elements, many unnecessary parts - commonly used measurement and calibration techniques, e.g. pH. It is not clear which microbiological analysis methods were used in addition to incubation. Absent information about used microbiological technigues. Statistical analyses were performed on actual data or transformed data? How were samples not containing the microbial groups tested avoided? On what basis was the CCA method chosen for the ordination analyses?
  • I do not understand the importance of describing the study area in the form of an independent point. What is the significance of providing information about the area.

Response: The reason why the authors chose to describe the study area (as well as provided adequate coordinate points) is to give the reviewers as well as everyone that come in contact with the paper upon publishing the chance to have an idea of the climatic and morphometric composition of the study area. It also proofs that the research is not fabricated and that it is original.

  • In addition, the inclusion of a third-level subsection "sampling collection" unnecessarily complicates the structure of the work.

Response: The phrase "Sampling collection" which implies the selection of the brands of sachet water that were analysed during the study has been rephrased to "Selection of samples for analysis"

  • The description of the procedure for the collection of empirical data contains, besides important elements, many unnecessary parts - commonly used measurement and calibration techniques, e.g., pH.

Response: The authors clearly stated the instrument used to analysed the water samples in the manuscript: Temperature, pH, Electrical conductivity and TDS of the sachet water were determined electrometrically with a multi-parameter data logger (Hanna model HI991300, Hanna Instruments, Singapore). However, the authors want clarity if the reviewer would prefer that the calibration description of the instrument be expunge from the write up?

  • Absent information about used microbiological techniques.

Response: The authors have included in detailed the microbiological techniques employed during the study as well as the media used and how bacterial and fungal isolates were identified.

Check lines 135 – 150

(e)      Statistical analyses were performed on actual data or transformed data?

             Response:  transformed data

(f)      How were samples not containing the microbial groups tested avoided?

Response:  I will employ the reviewer to check very well the figure 2 because the diagram contain       microbial group like total bacteria counts and total fungal counts correlate with floor storage A, B, Plank A, B, and C.

(g)      On what basis was the CCA method chosen for the ordination analyses?

          Response: It was PCA I used not CCA.

Results. Why are the tables and figures not in the correct places in the text? The form of presentation of the tabulated results is very poor (shifted lines) and difficult to verify. Significance levels of differences do not require additional explanations "*". Table titles do not retain the principle of fully explaining it content. What the letters A-C in Table 4 - stand for is not explained in the caption. The generic term "spp." or "sp." sholud be not in italics. In Figure 1, what do the lines on the bars mean - maybe "SD"? Figure 2 is PCA and the methodology does not describe that this technique was used. The graphic itself is truncated and illegible. There is no basic data on the quality of the model so it is difficult to determine whether an important results of the analysis is presented.

  • Why are the tables and figures not in the correct places in the text?

Response: Revised

  • The form of presentation of the tabulated results is very poor (shifted lines) and difficult to verify. Significance levels of differences do not require additional explanations "*".

Response: It has been adjusted and the asterisk sign has been removed

(c)      Table titles do not retain the principle of fully explaining it content.

          Response: Revised

(d)       What the letters A-C in Table 4 - stand for is not explained in the caption.

          Response: Revised, check lines 106 – 107. Alphabets A, B and C are the sample code to replace the real name for three brand of sachet water used for this research (A = TSK, B = M-    Square and C = Fahad).

(e)     The generic term "spp." or "sp." sholud be not in italics.

          Response: Revised

(f)      In Figure 1, what do the lines on the bars mean - maybe "SD"?

          Response: The line bars using in figure 1 are standard error bar

(g)      Figure 2 is PCA and the methodology does not describe that this technique was used. The graphic itself is truncated and illegible.

          Response: It has been corrected from CCA to PCA in the methodology. Check lines 155 -156.

(h)     There is no basic data on the quality of the model, so it is difficult to determine whether an important result of the analysis is presented.

          Response: The important results of these analysis is presented

Due to the number of ambiguities indicated, I have not analysed the discussion and conclusion in detail. Correcting the results will provide an opportunity to assess the authors' explanations of the observed patterns. 

Response: It has been revised

The validity and completeness of the literature was not checked.

Reviewer 2 Report

The work submitted for evaluation provides information on the physico-chemical and microbilogical composition of sachet waters consumed in a selected area of Nigeria. Given the problem of access to water in Africa, the idea of the thesis is valid. However, the form of the study is at a very poor level. This applies to both the technical and content layers. The methodological approach is presented in a very unclear manner. I do not know why water was considered to be stored in different places (apart from the fridge, of course). 

Abstract. Does not present the most important achievements of the work, but contains unexplained abbreviations and methodological elements. The authors need to consider in which mode they are writing: indicative or presumptive? The passage needs to be re-edited.

Keywords must not repeat elements of the title

Introduction. Passages on lines 77-79 indicate that the research presented is not new and is secondary to existing studies. Lines 82-83 contain content that has already been presented above in the body of the study. The main objective should be the verification of the hypothesis, obviously based on the information gained. This chapter needs to be rethought and rewritten.

Material and methods. I do not understand the importance of describing the study area in the form of an independent point. What is the significance of providing information about the area. In addition, the inclusion of a third-level subsection "sampling collection" unnecessarily complicates the structure of the work. The description of the procedure for the collection of empirical data contains, besides important elements, many unnecessary parts - commonly used measurement and calibration techniques, e.g. pH. It is not clear which microbiological analysis methods were used in addition to incubation. Absent information about used microbiological technigues. Statistical analyses were performed on actual data or transformed data? How were samples not containing the microbial groups tested avoided? On what basis was the CCA method chosen for the ordynation analyses?

Results. Why are the tables and figures not in the correct places in the text? The form of presentation of the tabulated results is very poor (shifted lines) and difficult to verify. Significance levels of differences do not require additional explanations "*". Table titles do not retain the principle of fully explaining it content. What the letters A-C in Table 4 - stand for is not explained in the caption. The generic term "spp." or "sp." sholud be not in italics. In Figure 1, what do the lines on the bars mean - maybe "SD"? Figure 2 is PCA and the methodology does not describe that this technique was used. The graphic itself is truncated and illegible. There is no basic data on the quality of the model so it is difficult to determine whether an important results of the analysis is presented.

Due to the number of ambiguities indicated, I have not analysed the discussion and conclusion in detail. Correcting the results will provide an opportunity to assess the authors' explanations of the observed patterns. 

The validity and completeness of the literature was not checked.

Author Response

Reviewer 2

The Introduction section is, in my opinion, too long; that's why I would recommend that this section be kept within the limits of the topic addressed.

Response: It has been revised

  1. The same observation in the case of the Materials and Methods section; the authors should retain the essential elements that ensure the reproducibility of the experiments

Response: It has been revised

  1. I also recommend the authors to check the way of writing the references (eg, APHA et al (2005) - line 23).

Response: corrected

Round 2

Reviewer 2 Report

Despite numerous revisions, the basic objections to the study have remained unchanged. The authors have changed the elements indicated in the study, but still do not fully realize who is right - the evaluator or them. This can be indicated in Figure 2. Once again, I analyzed the description under the graphic. In the review, I only suggested comparing the description of the methodology and the technique specified in the caption. Since they were discrepant I assumed that the description of the figure was correct. However, it turns out that it cannot be PCA in any case, because the authors combined different types of data (physicochemical parameters of water and biological parameters). The authors did either RDA or CCA analysis. Of course, I still don't know how it was done (on real or transformed data). The description of the ordination results is not quite correct either, as the authors only refer to correlations, but do not provide basic information about the model (e.g., F-ratio, p-value, level of explanation of selected axes). Overall, the description of the results is still weak.